# Recent Trend of Antimicrobial Susceptibility among *Mycoplasma pneumoniae* Isolated from Japanese Children

**DOI:** 10.3390/microorganisms10122428

**Published:** 2022-12-08

**Authors:** Tomohiro Oishi, Daisuke Yoshioka, Takashi Nakano, Kazunobu Ouchi

**Affiliations:** 1Department of Clinical Infectious Diseases, Kawasaki Medical School, Kurashiki 701-0192, Japan; 2Department of Pediatrics, Kawasaki Medical School, Kurashiki 701-0192, Japan; 3Department of Medical Welfare for Children, Kawasaki University of Medical Welfare, Kurashiki 701-0193, Japan

**Keywords:** antimicrobial resistance, minimum inhibitory concentration, *Mycoplasma pneumoniae*, pediatric pneumonia

## Abstract

Macrolide-resistant *Mycoplasma pneumoniae* (MRMP) infections have become increasingly prevalent, especially in East Asia. Whereas MRMP strains have point mutations that are implicated in conferring resistance, monitoring the antibiotic susceptibility of *M. pneumoniae* and identifying mutations in the resistant strains is crucial for effective disease management. Therefore, we investigated antimicrobial susceptibilities among *M. pneumoniae* isolates obtained from Japanese children since 2011. To establish the current susceptibility trend, we analyzed the minimum inhibitory concentrations (MICs) of *M. pneumoniae* in recent years (2017–2020) in comparison with past data. Our observation of 122 *M. pneumoniae* strains suggested that 76 were macrolide-susceptible *M. pneumoniae* (MSMP) and 46 were macrolide-resistant. The MIC ranges (µg/mL) of clarithromycin (CAM), azithromycin (AZM), tosufloxacin (TFLX), and minocycline (MINO) to all *M. pneumoniae* isolates were 0.001–>128, 0.00012–>128, 0.25–0.5, and 0.125–4 µg/mL, respectively. None of the strains was resistant to TFLX or MINO. The MIC distributions of CAM and AZM to MSMP and MINO to all *M. pneumoniae* isolates were significantly lower, but that of TFLX was significantly higher than that reported in all previous data concordant with the amount of recent antimicrobial use. Therefore, continuation of appropriate antimicrobial use for *M. pneumoniae* infection is important.

## 1. Introduction

*Mycoplasma pneumoniae* is a major pathogen that causes lower respiratory infections, mainly in children and youth [1], with antibiotics representing the main treatment option. Recently, macrolide-resistant *M. pneumoniae* [MRMP] strains have emerged in resistance to macrolides, a class of antibiotics that is commonly used to treat *M. pneumoniae* infections, especially in East Asian countries [2,3,4]. In these situations, alternative antibiotics, such as quinolone or tetracycline agents, should be considered. Approximately 0.5–2% of all *M. pneumoniae* pneumonia cases are the fulminant type, with a reported mortality rate or 3–5% in the 1980s [5]. The frequency of fulminant *M. pneumoniae* pneumonia due to MRMP is unclear. However, resistance causes difficulty in the management of *M. pneumoniae* infections. Therefore, the mortality of *M. pneumoniae* pneumonia due to MRMP is hypothesized to be not less than that of all *M. pneumoniae* pneumonia.

Therefore, monitoring the antibiotic susceptibility of *M. pneumoniae* is crucial. Multiple studies on MRMP in recent years [2,3,6,7,8,9,10,11,12,13,14] have identified point mutations in the V domain of the 23S rRNA sequence related to macrolides, such as the A2063G and A2064G transitions, using real-time PCR [15]. However, other mutations related to macrolide resistance and the occurrence of *M. pneumoniae* isolates that are resistant to alternative antibiotics cannot be ruled out. Therefore, the minimum inhibitory concentration (MIC) of *M. pneumoniae* in macrolides and other antibiotics must be determined.

In Japan, two epidemics of *M. pneumoniae* infections occurred in 2011–2012 and 2015–2016. However, no report on the MIC of *M. pneumoniae* was published after these pandemics.

Therefore, we aimed to investigate the antimicrobial susceptibility of *M. pneumoniae* isolates from Japanese children from 2017 to 2020, comparing recent data with those of past epidemics.

## 2. Materials and Methods

### 2.1. Sample Collection

All samples were collected from pediatric patients with acute respiratory tract infections at 85 institutions located in eight areas throughout Japan (20 institutions in Kyushu, 25 in Chugoku, 3 in Shikoku, 11 in Kinki, 7 in Chubu, 3 in Kanto, 2 in Tohoku, and 3 in Hokkaido) from 2011 to 2020. The study protocol was approved by the Ethics Committee of Kawasaki Medical School, Kurashiki, Japan, on 8 September 2021 (no. 3119-04), and we obtained parental consents for this study.

### 2.2. M. pneumoniae Isolation

*M. pneumoniae* isolates were obtained by specimen cultivation. Pleuropneumonia-like organism broth (PPLO) (Oxoid, Hampshire, UK) supplemented with 0.5% glucose (FUJIFILM Wako Pure Chemical Corporation, Osaka, Japan), 20% mycoplasma supplement G (Oxoid), and 0.0025% phenol red (Sigma-Aldrich, St. Louis, MO, USA) was used for isolation and MIC determination for selection of only *M. pneumoniae* as described in [16].

### 2.3. Antimicrobial Susceptibility Testing

The MICs of antimicrobial agents for the isolated strains were determined using microdilution methods [17]. Medium containing 10^5^–10^6^ CFU/mL *M. pneumoniae* was added to 96-well microplates and incubated at 37 °C for 6–8 days. The MIC was defined as the lowest concentration of antimicrobial agent at which the metabolism of the organism was inhibited, which was evidenced by a lack of color change in the medium three days after the drug-free control first exhibited color change. The reference strain, FH, was used as a drug-susceptible control. Clarithromycin (CAM), azithromycin (AZM), tosufloxacin (TFLX), and minocycline (MINO) were the antimicrobial agents used for MIC determination. Each antibiotic concentration was set from 0.000013 to 128 μg/mL as described in [16].

### 2.4. Statistical Methods

GraphPad Prism 5 (GraphPad Software Inc., San Diego, CA, USA) was used for statistical analysis. Differences between the two groups were analyzed using the chi-squared test, Student’s *t*-test, Fisher’s exact text, or Mann–Whitney U test, and the 95% confidence interval was determined. Results are expressed as mean ± standard deviation (SD). *p* values of <0.05 were considered significant.

## 3. Results

### 3.1. In Vitro Antimicrobial Activity

Table 1 lists the last four years of data representing the in vitro antimicrobial activity of the selected agents for the treatment of *M. pneumoniae* infections from 2017 to 2020.

The MIC ranges of two macrolide agents, CAM and AZM, for all *M. pneumoniae* were notably large, and the MIC_50_ and MIC_90_ values for macrolide-susceptible *M. pneumoniae* (MSMP) were notably low. However, these values were high for MRMP. The MIC ranges of the quinolone and tetracycline agents TFLX and MINO were relatively small. Furthermore, the MIC_50_ and MIC_90_ values of these two agents regarding MRMP and MSMP were near identical.

### 3.2. MIC Distribution of Macrolide Agents against M. pneumoniae Isolates during Three Time Periods

Figure 1 shows the cumulative distribution of the MICs of two macrolide agents, CAM and AZM, during three periods: the first recent epidemic of 2011 and 2012, the second recent epidemic of 2015 and 2016, and the most recent epidemic from 2017 to 2020.

As depicted in Figure 1, all isolates of *M. pneumoniae* tested against both macrolide agents were grouped into two masses. The masses on the left and right sides predominantly represent MSMP and MRMP groups, respectively. In the MSMP group, the MICs of CAM and AZM were significantly lower during the last few years than during the first recent epidemic.

### 3.3. The MIC Distribution of TFLX against M. pneumoniae Isolates during Three Time Periods

The cumulative MIC distribution of TFLX is indicated in Figure 2. The TFLX MICs of the isolates were significantly higher in recent years than during the first and second epidemics (*p* < 0.0001). However, all TFLX MICs were grouped under a single mass, as shown in Figure 2. Accordingly, no resistant isolates against TFLX were detected during the investigated periods.

### 3.4. MIC Distribution of MINO against M. pneumoniae Isolates during Three Time Periods

Figure 3 represents the cumulative MIC distribution of MINO; the MICs of MINO during the second and third epidemics were significantly lower than those observed during the first recent epidemic (*p* < 0.0001). All MICs of MINO against isolates belonged to a single mass, suggesting that no MINO-resistant isolate was detected in recent years.

## 4. Discussion

In the MIC distribution of antimicrobial agents, the macrolide MICs of MSSP were significantly lower during the last few years than during the first recent epidemic. The TFLX MICs were significantly higher during the third epidemic than in the first and second recent epidemics. Finally, the MICs of MINO were significantly lower during the third epidemic than during the first and second recent epidemics. These results have two explanations. First, the Japanese guidelines for *M. pneumoniae* infections were published in 2014 [18]. Specifically, these guidelines state that macrolides are recommended as the first-line drug of choice for treatment of *M. pneumoniae* infections. The macrolide efficacy has a relatively high accuracy in the presence or absence of defervescence within 48–72 h of initiating macrolide treatment. Second, the use of TFLX or tetracyclines may be considered when required for patients with pneumonia who do not respond to macrolides. Therefore, we assume that clinicians prescribed antimicrobial agents appropriately, that is, they did not continue to prescribe macrolides for insensitive infections, and antimicrobial agents other than macrolides were used more often. Okubo Y et al. (2018) reported that the use of macrolide agents for pediatric *M. pneumoniae* infections has recently decreased, whereas that of quinolone agents has increased [19], suggesting that the amount of antibiotics used by clinicians is influenced the MIC change. Future considerations include rapid diagnosis kits to detect *M. pneumoniae* antigens and other factors such as point mutations that confer macrolide resistance [20,21]. Clinicians have been able to immediately identify whether patients with *M. pneumoniae* infections harbor macrolide-resistant strains. Therefore, the advent of these diagnostic kits has seemingly facilitated appropriate antimicrobial usage. However, tetracyclines such as MINO are contraindicated in children younger than eight years of age per the Japanese guidelines [16], and the average age of patients with *Mycoplasma* infections is approximately six years. Thus, many suspected MRMP cases have been prescribed TFLX instead of MINO. Furthermore, pediatricians in Japan frequently prescribe TFLX rather than MINO, which is a common practice.

The MIC distribution of TFLX is higher than that in the past; however, it has been approved for treatment of children with *M. pneumoniae* infection in Japan since 2010 and has played an important role against MRMP infections. Several patients with infections have been cured promptly by TFLX, and its growing use has suppressed the occurrence of MRMP. Notably, the rate of MRMP among Japanese children has decreased in recent years [12]. Ouchi et al. (2017) reported the clinical effectiveness and efficient eradication rates (including MRMP) of TFLX [22]. Thus, MRMP was effectively inhibited by TFLX, consecutively lowering the rate of MRMP. Among *M. pneumoniae* isolates, no resistant strains were identified against TFLX. Therefore, TFLX must be continually prescribed to effectively combat *M. pneumoniae* infections. 

This study is subject to some limitations. First, we did not analyze the backgrounds of children affected by *M. pneumoniae*. However, we collected many samples throughout Japan, which may minimize the magnitude of differences among the samples. Second, we only examined the MICs of antimicrobial agents but did not analyze other factors such as genetics and molecular epidemiology. Therefore, in future studies, our analysis should be broadened to include such factors. 

In conclusion, we investigated the MICs of antibiotics against *M. pneumoniae* isolated from Japanese children and MIC distributions of macrolide agents. We found that MINO MICs are lower, whereas those of TFLX are higher than those in the past, in accordance with the increased usage of these drugs. We did not identify quinolone- or tetracycline-resistant *M. pneumoniae;* however, constant surveillance is required in the future.

## Figures and Tables

**Figure 1 microorganisms-10-02428-f001:**
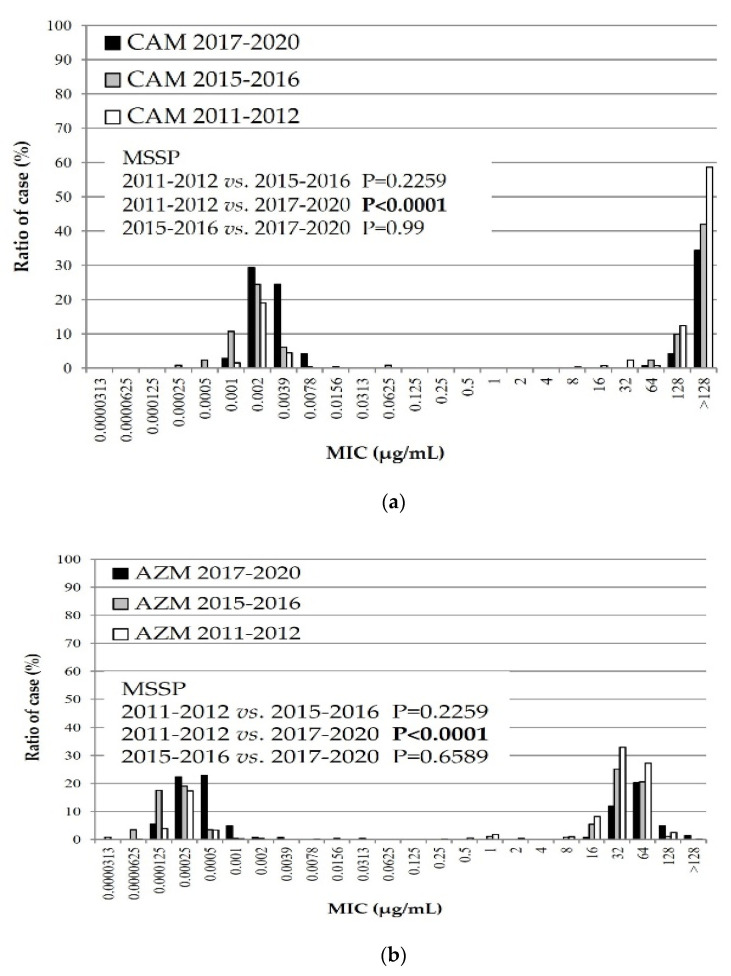
Minimum inhibitory concentration (MIC) distribution of macrolide agents, (**a**) CAM and (**b**) AZM, against *Mycoplasma pneumoniae* isolates during three periods: 2011–2012, 2015–2016, and 2017–2020. CAM: clarithromycin, AZM: azithromycin.

**Figure 2 microorganisms-10-02428-f002:**
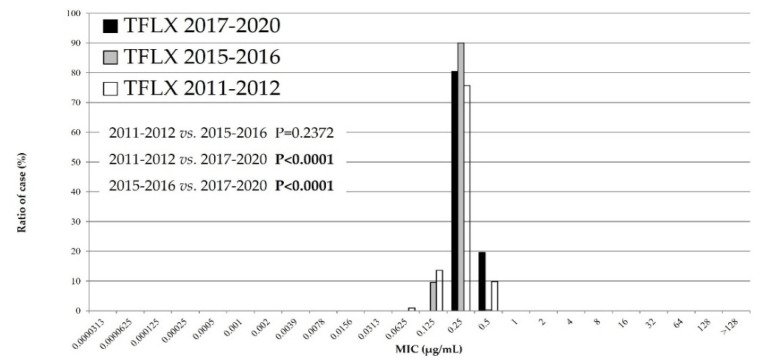
Minimum inhibitory concentration (MIC) distribution of tosufloxacin against *Mycoplasma pneumoniae* isolates during three periods: 2011–2012, 2015–2016, and 2017–2020. TFLX: tosufloxacin.

**Figure 3 microorganisms-10-02428-f003:**
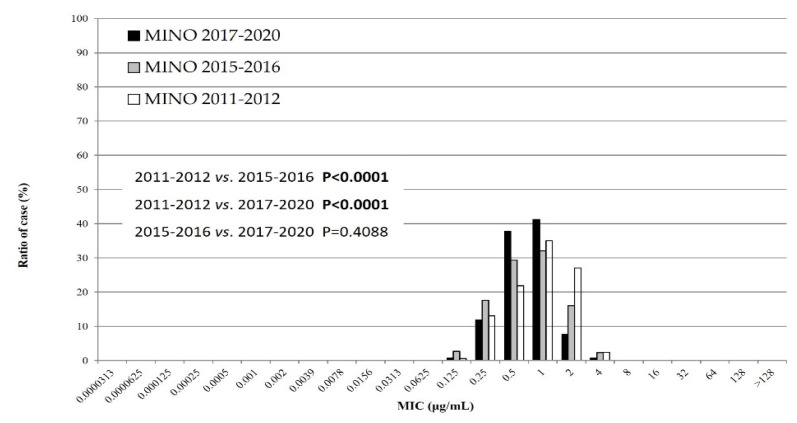
MIC distribution of minocycline against *M. pneumoniae* isolates during the three periods: 2011–2012, 2015–2016, and 2017–2020. MINO: minocycline.

**Table 1 microorganisms-10-02428-t001:** In vitro antimicrobial activity against clinical isolates of *Mycoplasma pneumoniae* strains from 2017 to 2020. Antimicrobial susceptibility testing was performed as reported in [16].

Organism (Number of Strains) (n = 122)	Antimicrobial Agents	MIC (μg/mL)				
		MIC Range			MIC_50_	MIC_90_
** *Mycoplasma pneumoniae* ** **(122)**	CAM	0.001	–	>128	0.0039	>128
	AZM	0.00012		>128	0.0005	64
	TFLX	0.25	–	0.5	0.25	0.5
	MINO	0.125	–	4	0.5	1
**Macrolide-susceptible** ***M. pneumoniae* (76)**	CAM	0.0078	–	0.001	0.002	0.0039
	AZM	0.00012		0.0039	0.0005	0.001
	TFLX	0.25	–	0.5	0.25	0.5
	MINO	0.25	–	4	1	1
**Macrolide-resistant** ***M. pneumoniae* (46)**	CAM	16	–	>128	>128	>128
	AZM	64		>128	64	128
	TFLX	0.25	–	0.5	0.25	0.25
	MINO	0.125	–	2	0.5	2

TFLX: tosufloxacin, MINO: minocycline, CAM: clarithromycin, AZM: azithromycin; MIC: minimum inhibitory concentration.

## Data Availability

Not applicable.

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
