# Peer review of "Recent Trend of Antimicrobial Susceptibility among Mycoplasma pneumoniae Isolated from Japanese Children"

_microorganisms, 2022, doi:10.3390/microorganisms10122428_

Round 1
Reviewer 1 Report
General:
Oishi T et al. concisely described in this paper the susceptibility profiles of some antimicrobials which are commonly used for the treatment of M. pneumonia infection with special reference to the time period of sample isolation. The study is well constructed and the style of presentation is appropriate.
Specific:
Abstract: This reviewer feels that the abstract ends rather abruptly stating up to results without conclusion. The authors are encouraged to close the abstract with some conclusive words for stating the most important message that the authors would convey to the readers through this article.
Author Response
Response to Reviewer 1 Comments
Thank you for your kind comments.
I revised as you indicated. I described as below.
Abstract: This reviewer feels that the abstract ends rather abruptly stating up to results without conclusion. The authors are encouraged to close the abstract with some conclusive words for stating the most important message that the authors would convey to the readers through this article.
Thank you for your suggestion. I revised the abstract on the line 22-23, and 25-26.

Reviewer 2 Report
The article is very well written, and its experimental part is very well reported.
Some considerations:
- The introduction must be supplemented. Bring more data on the disease caused by the bacteria, data on deaths related to resistant pathogens, among other relevant information.
- In methodology section, was there confirmation of the species of the isolates? Is the method used sufficient to guarantee that only M. pneumoniae were isolated?
- The MIC method should be better detailed. What antibiotic concentrations were used? What temperature was used? Why 6-8 days of incubation, not just 24 hours? This information should be added, as well as any other relevant information.
- In the methodology, the previous data collection method should be added. In addition, appropriate references must be added in Table 1.
- In results, the images are too small. I suggest that, instead of being horizontal, they are vertical, so that their size can be increased.
- To improve the discussion, I suggest adding quantitative data on the amount of antibiotics used in the treatment. Are these data available?
Author Response
Response to Reviewer 2 Comments
Thank you for your kind comments.
I revised as you indicated. I described as below.
- The introduction must be supplemented. Bring more data on the disease caused by the bacteria, data on deaths related to resistant pathogens, among other relevant information.
Thank you for your indication. I added the sentence and the reference on the line 36-39 and 40-42.
- In methodology section, was there confirmation of the species of the isolates? Is the method used sufficient to guarantee that only M. pneumoniae were isolated?
Thank you for your questions. I wrote the sentence and the reference on the line 67.
- The MIC method should be better detailed. What antibiotic concentrations were used? What temperature was used? Why 6-8 days of incubation, not just 24 hours? This information should be added, as well as any other relevant information.
Thank you for your questions. I added the antibiotic concentrations and the reference in line 77-78. Then, the reason for incubation time is that the culture time for M. pneumoniae is more than 7 days, and was based on the reference.
- In the methodology, the previous data collection method should be added. In addition, appropriate references must be added in Table 1.
Thank you for your suggestion. I added the sentence and the reference on the line 90.
- In results, the images are too small. I suggest that, instead of being horizontal, they are vertical, so that their size can be increased.
Thank you for your suggestion. I enlarged the Figure 1(a) and (b) to horizontal direction.
- To improve the discussion, I suggest adding quantitative data on the amount of antibiotics used in the treatment. Are these data available?
Thank you for your suggestion and question. I enlarged the Figure 1(a) and (b) to horizontal direction. I added the comment based on the reference 21 on the line 146-147.
